# Who participates in 'participatory design' of WASH infrastructure: A mixed-methods process evaluation

**Thea L. Mink**[1], **Allison P. Salinger**[1], **Naomi Francis**[2], **Becky Batagol**[2,3], **Kerrie Burge**[2], **Noor Ilhamsyah**[2,4], **Losalini Malumu**[2,5], **Liza Marzaman**[2,4], **Michaela F. Prescott**[6], **Nur Intan Putri**[2,4], **Sheela S. Sinharoy**[1]*

**1** Hubert Department of Global Health, Rollins School of Public Health, Emory University, Atlanta, Georgia, United States of America, **2** Monash Sustainable Development Institute, Monash University, Melbourne, Australia, **3** Faculty of Law, Monash University, Melbourne, Australia, **4** Indonesia Team, Revitalising Informal Settlements and their Environments (RISE), Makassar, Indonesia, **5** Fiji Team, Revitalising Informal Settlements and their Environments (RISE), Suva, Fiji, **6** Faculty of Art, Design and Architecture, Monash University, Melbourne, Australia

* sheela.sinharoy@emory.edu

## Abstract

Inclusive participation is critical for community-based water, sanitation and hygiene (WASH) interventions, especially in complex environments such as urban informal settlements. We conducted a mixed-methods, theory-driven process evaluation to evaluate participation, barriers to participation, and participant satisfaction within the Revitalising Informal Settlements and their Environments (RISE) trial in Makassar, Indonesia and Suva, Fiji (ACTRN12618000633280; https://www.anzctr.org.au/). RISE conducted participatory design activities, including community-level design workshops and household visits, to co-design WASH infrastructure. Household surveys, conducted with women and men after RISE participatory design in Makassar (N = 320) and Suva (N = 503), captured self-reported participation in RISE activities and satisfaction with influence over RISE decision-making. We used logistic regression models to assess socio-demographic predictors of participation and satisfaction. Qualitative data were also collected after participatory design and analyzed thematically. Most respondents from Makassar (89%) and Suva (75%) participated in at least one RISE activity. Statistically significant predictors of participation included gender, age, and marital status in Makassar and disability status and education in Suva. Most participants in Makassar (66%) and Suva (70%) were satisfied with their level of influence over RISE decision-making. In Makassar, no significant predictors of satisfaction were identified. In Suva, significant predictors of satisfaction included gender, religion, and marital status, with women wanting *more* influence and religious minorities and unmarried participants wanting *less* influence over decisions. Qualitative data showed that most participants felt satisfied by RISE's inclusive and participatory design, although some residents reported distrust with RISE and feeling excluded from activities by

**Data availability statement:** Data are available via FigShare: DOI 10.6084/ m9.figshare.c.7645769.

**Funding:** The data collected for this project was made possible by the generous support of the Australian Department of Foreign Affairs and Trade (DFAT) through the Water for Women Fund [WRA 1023 to BB and SS]. The RISE program is funded by the Wellcome Trust [grant 205222/Z/16/Z], the New Zealand Ministry of Foreign Affairs and Trade, the Australian Department of Foreign Affairs and Trade, the Government of Fiji, the Asian Development Bank and Monash University, and involves partnerships and in-kind contributions from the City of Makassar, the Cooperative Research Centre for Water Sensitive Cities (now Water Sensitive Cities Australia), Fiji National University, Hasanuddin University, Stanford University, Emory University, Melbourne University, Southeast Water, Melbourne Water, Live and Learn Environmental Education, UN-Habitat, UNU-IIGH, WaterAid International and Oxfam. The funders had no role in study design, data collection and analysis, decision to publish, or preparation of the manuscript.

**Competing interests:** The authors have declared that no competing interests exist.

community representatives. While RISE participatory design activities achieved good reach and satisfaction overall, we identified specific gender and social inequities in participation and influence over decision-making. We recommend that WASH interventions reflect on the quality of their engagement with communities and local organizations in order to identify and appropriately include groups of interest.

## 1 Introduction

1.1 billion people live in urban informal settlements and regularly contend with unimproved water and sanitation services (including poor access, quality, and reliability), in addition to insufficient living areas, poor housing durability, and unstable land tenure [1–3]. Marginalized groups living in urban informal settlements, like women, the elderly, and people living with disabilities, are often at the intersection of multiple deprivations, experiencing social exclusion in addition to poor water and sanitation conditions [4–9]. Sustainable Development Goal (SDG) 11 includes target 11.1 to, "Ensure access for all to adequate, safe and affordable housing and basic services and upgrade slums" by 2030 [10]. Ensuring basic services is further promoted by SDG 6, which aims to provide access to water and sanitation for all populations, with target 6.2 further specifying, "Paying special attention to the needs of women and girls and those in vulnerable situations" [10]. To achieve these SDGs in complex environments like urban informal settlements, comprehensive water and sanitation interventions are needed [11–13].

Community participation is a central part of comprehensive water and sanitation interventions. Participatory approaches are used to tailor interventions to communities' environmental, cultural, and economic landscapes, and preferences [14]. Community participation can contribute to benefits like increased intervention awareness and acceptance, enhanced community ownership, and improved management and sustainability [15]. Deliberate inclusion of marginalized people can help ensure that their specific needs are incorporated into water and sanitation programming and confer additional benefits for those involved. For example, inclusive WASH programs have bolstered women's confidence and community respect as decision-makers [16,17].

Revitalising Informal Settlements and their Environments (RISE) is a randomized control trial aiming to reduce environmental contamination and improve human and ecological health through a water-sensitive infrastructure intervention in Suva, Fiji and Makassar, Indonesia [18]. In each country, 12 urban informal settlements were selected, of which six settlements were randomly allocated to receive the RISE intervention, with the remaining six control settlements to be offered the intervention at a later date [18]. The RISE intervention includes nature-based and decentralized components (pressure tanks, constructed wetlands, and rainwater tanks) to decrease exposure to fecal contamination [18]. These components were selected because conventional water and sewer systems do not often service informal settlements, and the evidence is mixed for other lower-cost water and sanitation interventions [18–25].

Community engagement activities were a key component of the RISE infrastructure design and planning process for each intervention settlement [13,18]. Specifically,

participatory design workshops and household visits were planned to reach a diverse population (with a particular focus on marginalized groups and women) and to meaningfully collaborate with participants in intervention decision-making [26,27]. The participatory design activities helped identify and adapt to residents' needs, which included solutions to environmental stressors, and residents were motivated to participate because of RISE's emphasis on inclusion [27]. Participatory design workshops included mapping of important community features, discussions of upgrading design options, and marking out potential infrastructure locations [26]. Household visits discussed individual household conditions and preferences for infrastructure connection [26]. Further information about the RISE program and participatory design process is outlined elsewhere [18,26,27].

Along with the movement towards comprehensive WASH interventions, there are calls for more comprehensive evaluation of such interventions [12,13,28,29]. Towards that goal, we conducted two companion studies relating to the RISE participatory design phase: one on implementation science mechanisms for inclusion and the present process evaluation [27]. Process evaluations, in particular, are rarely used to examine WASH interventions [28,30]. Process evaluations can assess if interventions were designed and implemented as planned and help distinguish between intended and delivered program components [30]. Process evaluations also provide context for program development and scaling up [31]. Among the several water and sanitation process evaluations in the literature, only a few have identified factors that affected participation and assessed participant satisfaction with interventions [6,32–34]. Notably, there is little published literature on process evaluations of the design phase of WASH interventions.

Saunders et al.'s process evaluation framework provides a systematic guide to assess the implementation of health interventions [30]. The framework's process indicators include reach (proportion of target audience that participates) and dose received (extent of participant engagement and satisfaction with an intervention) among others [30,31]. Reach and dose received can help identify whether certain groups are over or under-represented among a population of interest and assess intervention quality and acceptability from the perspective of participants [30,31].

We conducted a mixed-methods process evaluation to assess participation and barriers to participation (reach), and participant satisfaction with engagement (dose received) with a focus on gender and social inclusion for RISE participatory design activities. Saunders et al.'s process evaluation framework guided our analysis [30].

Our process evaluation had the following research questions:

1. Who participated in the primary RISE participatory design activities? (reach)

    a. What were the socio-demographic predictors of participation in the RISE participatory design workshops and household visits?

    b. What were the main barriers to participation in RISE activities?

2. What were participants' satisfaction with engagement in RISE activities? (dose received)

    a. What were the socio-demographic predictors of satisfaction among participants of RISE participatory design workshops and household visits?

    b. What were participants' opinions of RISE and engagement?

Findings from this process evaluation will help assess the implementation of RISE participatory design activities, as well as facilitate understanding of how process evaluations and participatory design activities can be better implemented in future water and sanitation interventions.

## 2 Methods

### 2.1 Ethics statement

The data analyzed in this study were collected in accordance with relevant guidelines and regulations as part of the RISE trial, which is registered with the Australian and New Zealand Clinical Trials Registry (ACTRN12618000633280;

https://www.anzctr.org.au/). Ethics approval for the RISE trial and this sub-study were obtained from the Monash University Human Research Ethics Committee (Melbourne, Australia; project IDs 35903 and 22726) effective as of November 2, 2022, Universitas Hasanuddin (Makassar, Indonesia; protocols UH18020110 and UH20050235), and Fiji National University (Suva, Fiji; protocol 137.19). Additional approval for this study was obtained from the University of the South Pacific. The parent study included Emory University researchers in IRB applications prior to data collection. Heads of households provided written informed consent for participation of their household members in the RISE trial at baseline, which took place in late 2018 in Makassar and mid-2019 in Suva [18]. RISE staff provided written informed consent for participation in IDIs.

## 2.2 Study design

Within the RISE trial, informal settlements in urban areas of Makassar, Indonesia and Suva, Fiji were purposively selected for inclusion based on community willingness and commitment to participate; settlement size, location, and demographics; and environmental and construction-related conditions [18]. Covariate-constrained randomization was then used to allocate settlements to the intervention and control groups [18]. RISE attempted to enroll all households in each settlement, and enrollment and baseline surveys took place in late 2018 in Makassar and mid-2019 in Suva [18]. Participatory design activities, conducted by RISE staff, commenced soon after the baseline surveys in each country and continued through October 2019 in Makassar and December 2020 in Suva (Fig 1). All data collection for this process evaluation was conducted after completion of participatory design and community engagement activities, but before the construction of physical infrastructure in intervention settlements.

For the purpose of this process evaluation, we used both quantitative and qualitative methods to assess reach and dose received for the participatory community engagement activities in RISE. We operationalized reach as the proportion of quantitative survey respondents reporting participation in two primary RISE participatory design activities (community-level participatory design workshops and household visits). Dose received was operationalized in the

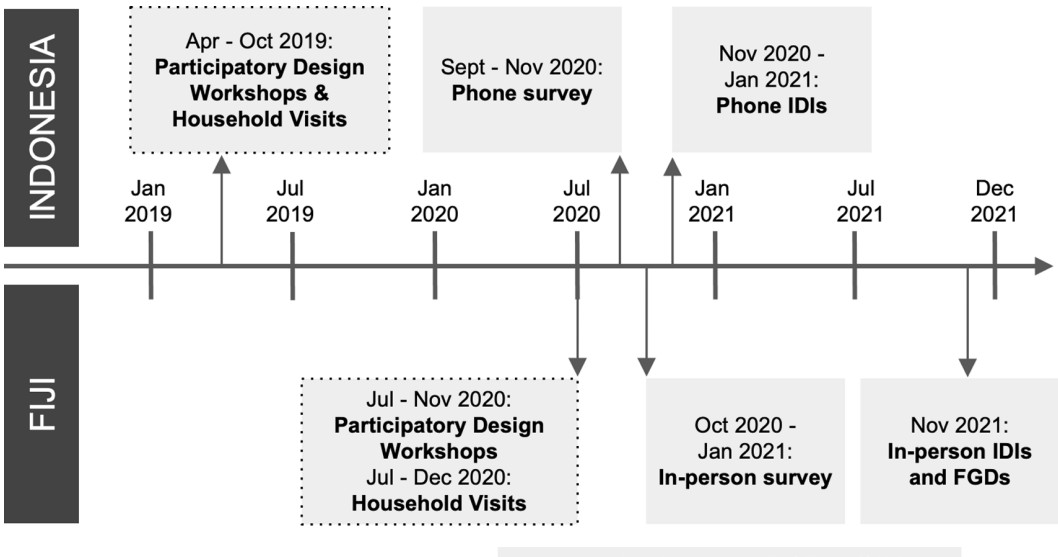

**Fig 1. Timeline of primary RISE participatory design activities (participatory design workshops and household visits) and data collection activities.**

quantitative work as participant-reported satisfaction during the two primary participatory design activities and in the qualitative work as participant opinion of overall RISE activities.

## 2.3 Quantitative methods

### 2.3.a Participants and procedures.

This process evaluation used quantitative data from surveys that targeted all intervention and control households that had previously enrolled and consented to RISE in Makassar (11 settlements, 593 households) and Suva (12 settlements, 767 households). Two adult survey respondents, one man and one woman, were targeted from each household. The survey tool was translated from English into local languages (Bahasa Indonesia for Makassar and iTaukei and Fijian Hindi for Suva) and then back-translated. The tool was then pilot tested in mock surveys in both countries prior to deployment [35]. In Makassar, trained local enumerator teams collected all survey data by phone because of COVID-19 restrictions. Data collection took place from September 23, 2020 to November 23, 2020 and was done in Bahasa Indonesia; verbal translation into Makassarese and Buginese was conducted with participants as needed. In Suva, local field teams administered surveys in-person from October 27, 2020 to January 28, 2021 in iTaukei and Fijian Hindi as appropriate. All teams collected data on tablets equipped with the SurveyCTO application. Per RISE protocols, individuals were considered a non-response for the survey if they were not reached after three phone calls in Makassar and after two house visits in Suva. Further information on the methods for the quantitative survey design and administration can be found elsewhere [35].

The survey tool included sections relating to self-reported participation in various RISE activities and perceived influence in program-related decision-making (see S1 Table for survey questions and response options). Participation-related questions asked respondents which RISE activities they participated in and their barriers to participation if they did not participate in any RISE activities. Data from these questions were used to assess 'reach' of the RISE activities. Questions on satisfaction asked how much influence respondents and their households felt they had over RISE decision-making, along with how much influence respondents and their households would have preferred. Satisfaction questions were asked regardless of whether the respondent had participated in RISE activities or not. Data from these questions were used to assess 'dose received.'

Respondents' disability status (including vision, hearing, mobility, cognition, self-care, and communication difficulties) was assessed using the Washington Group on Disability Statistics' short set of questions on functioning [36]. Other participant socio-demographic data that were used for analysis (gender, age, education level, marital status, ethnicity, religion, and asset ownership) were collected from earlier RISE surveys.

### 2.3.b Quantitative data analysis.

Our analytic sample included data from intervention settlements only. Analysis was conducted separately by country because of differences in intervention timelines and socio-cultural contexts. Two activities (participatory design workshops and household visits) were selected as the primary focus for analysis, because they were the main components of RISE's participatory design phase in both countries [26,27]. Other RISE activities (e.g., household data collection activities, trial consenting visits, and randomization workshops) were considered secondary activities for the purpose of this analysis.

Participation for all RISE activities and for primary participatory design activities was determined by calculating binary (yes/no) response frequencies. We identified gender, education, disability, ethnicity, religion, marital status, age, and asset ownership score *a priori* as possible predictors of participation. Logistic regression models were fit to examine bivariate associations between each predictor and participation in participatory design workshops or, separately, household visits. Predictor variables were selected for inclusion in full models if their coefficient p-value was less than 0.25 [37]. Multi-variable logistic regression models were then used to determine associations between socio-demographic predictors and participation in participatory design workshops and, separately, household visits. The full models were adjusted for settlement-level clustering.

Frequencies of barriers to participation were calculated for respondents who reported that they did not participate in any RISE activity. Because of the small sample size, we did not conduct any type of inference testing on the sub-sample who did not participate in any RISE activities.

Satisfaction with RISE decision-making was determined by comparing participants' experienced level of influence (none, a little, a lot) to participants' preferred level of influence (none, a little, a lot). Satisfaction was calculated only for participants who reported that they had participated in at least one of the primary RISE participatory design activities. Those whose experienced and preferred levels of influence matched were categorized as *satisfied* with their program influence. In contrast, those whose experienced and preferred levels of influence did not match were categorized as *discordant*. Logistic regression was used to determine the odds of influence discordance (wanting more influence vs. satisfied and wanting less influence vs. satisfied) over RISE-related decision-making for socio-demographic predictors. The same logistic regression methodology for the participation analysis was applied to this analysis.

All quantitative analyses were performed in Stata (version 17.0). We discussed the quantitative results with Makassar and Suva RISE staff during two contextualization workshops to further understand and validate the findings, an equitable authorship process recommended by Sam-Agudu and Abimbola [38].

## 2.4 Qualitative methods

**2.4.a Participants and procedures.** Qualitative data collection included semi-structured in-depth interviews (IDIs) with RISE staff and settlement residents in both countries and focus group discussions (FGDs) with residents in Fiji. Staff were purposively selected to participate in IDIs to maximize variation in country (Fiji, Indonesia, Australia, and the United States), staff role, and gender [27]. IDIs were conducted with a total of 49 RISE staff, over Zoom, between April 2020 and May 2021.

In Makassar, 17 semi-structured IDIs with residents took place by phone from November 2020 to January 2021. Residents were purposively selected for IDIs to maximize variation in gender, disability status, and extent of participation in RISE-related activities (determined from the quantitative survey) [27]. Data from all of these IDIs were included in the final analysis. FGDs were not performed in Makassar due to COVID-19 conditions [27].

In Suva, FGDs were facilitated in November 2021 and were composed of one women's group and one men's group per intervention site. Community liaisons suggested potential participants based on who had attended the most participatory design and engagement activities [27]. The final analysis includes data from six FGDs: the men's and women's FGD from three of the settlements (total of 48 participants); these were selected by the Fiji data collection team based on the depth of the participants' responses [27].

Additionally in Suva, 25 semi-structured IDIs were conducted in November 2021. The sampling frame consisted of residents who had been identified by FGD participants as potentially marginalized within their communities [27]. Data from 12 of the Fiji IDIs were selected: one man and one woman from each of the settlements (bar one where only women were interviewed).

In Makassar, semi-structured IDI and FGD guides were translated from English to Bahasa Indonesia and independently back-translated; questions were verbally translated into Makassarese or Buginese as needed. In Fiji, the English guides were translated verbally into iTaukei or Fiji Hindi based on participant preference. The IDI and FGD guides focused on RISE engagement and participation, in addition to community needs and resources, and were based in part on the Consolidated Framework for Implementation Research (CFIR) [27,39].

Further information on the qualitative methods can be found elsewhere [27,35].

**2.4.b Qualitative data analysis.** The qualitative data for this study was analyzed thematically in MAXQDA (VERBI Software, 2021). The codebook was drafted based on the CFIR (which had been contextualized for the RISE program context) and then refined through a memoing process until intercoder agreement (checked qualitatively and quantitatively) was reached among three researchers [40,41]. The 'full' codebook was used for staff data and a smaller 'subset' of the codebook was used for resident data [27].

The coded segments (which corresponded to the CFIR constructs) were exported and organized by participant group. The coded segments were then thematically analyzed for barriers to participation and satisfaction with RISE engagement. As with the quantitative results, the qualitative findings were discussed during contextualization workshops with RISE staff as an additional validation step.

## 4 Results

The response rate for the survey was 66% in Makassar (1,172 targeted) and 92% in Suva (1,308 targeted). Our analytic sample, which included only those in intervention settlements, consisted of 320 surveys (185 with women and 135 with men) from five sites in Makassar and 503 surveys (254 with women and 249 with men) from six sites in Suva. In both cities, a majority of participants had a secondary education or higher, were married, and were of the predominant ethnicity and religion. Additional socio-demographic characteristics of quantitative survey participants are shown in Table 1. The characteristics of qualitative participants are included in S2 Table.

### 4.1 Who participated in RISE activities? (reach)

Of the 319 survey respondents in Makassar for whom we had participation data, 283 (89%) participated in at least one RISE activity (including primary and secondary participatory activities). Among the 260 respondents who participated in at least one primary participatory activity, 200 (77%) participated in both community workshops and household visits, 31 (12%) participated in just household visits, and 29 (11%) participated in just community workshops.

Of the 500 Suva respondents for whom we had participation data, 375 (75%) participated in at least one RISE activity. Among the 301 respondents who participated in at least one primary participatory activity, 196 (65%) participated in both participatory design workshops and household visits, 99 (33%) participated in just participatory design workshops, and 6 (2%) participated in just household visits.

Socio-demographic characteristics varied by country and participatory activity and are outlined in S3 Table.

### 4.2 What were the predictors of participation in RISE participatory activities? (reach)

For Makassar participatory design workshops, the odds of participation were 2.6 times higher for women compared to men (95% confidence interval (CI): 1.8, 3.8) and 2.1 times higher for married respondents compared to respondents with other marital statuses (95% CI: 1.3, 3.4). For every one year increase in age in years, the adjusted odds of participation increased by 3% (95% CI: 1.02, 1.03). Ethnicity was not a significant predictor of participation in Makassar participatory design workshops (Table 2).

For Makassar household visits, the odds of participation were 3.6 times higher for women compared to men (95% CI: 1.4, 9.5). For every one year increase in age in years, the adjusted odds of participation increased by 5% (95% CI: 1.0, 1.1) (Table 2). Marital status was not a significant predictor of participation in Makassar household visits.

For Suva participatory design workshops, the odds of participation were 92% lower for respondents living with disabilities compared to respondents living without disabilities (95% CI: 0.0, 0.8) (Table 2).

For Suva household visits, the odds of participation were 1.7 times higher for people who completed secondary education or above compared to those who completed primary education or below (95% CI: 1.2, 2.5). Disability was not a significant predictor of participation in Suva household visits (Table 2).

### 4.3 What were the main barriers to participation in RISE activities? (reach)

Of the 36 Makassar respondents and 125 Suva respondents who did not participate in any RISE activity, the majority reported that they were too busy with work, housework, or school to participate (81% in Makassar, 86% in Suva) (Table 3).

**4.3.a Barriers related to RISE program delivery.** The qualitative data indicated that participants had a variety of challenges for participation in RISE activities. Around half of the Makassar resident participants (with high and low

**Table 1. Socio-demographic characteristics of Makassar and Suva survey respondents.**

| Variables | Makassar (n = 320) | | Suva (n = 503) | |
|---|---|---|---|---|
| | n | (%) | n | (%) |
| **Gender** | | | | |
| Women | 185 | (57.8%) | 254 | (50.5%) |
| Men | 135 | (42.2%) | 249 | (49.5%) |
| Total | 320 | (100.0%) | 503 | (100.0%) |
| **Education** | | | | |
| Primary & below | 137 | (43.2%) | 78 | (17.4%) |
| Secondary & above | 180 | (56.8%) | 371 | (82.6%) |
| Total | 317 | (100.0%) | 449 | (100.0%) |
| **Disability** | | | | |
| No | 305 | (95.3%) | 493 | (98.2%) |
| Yes | 15 | (4.7%) | 9 | (1.8%) |
| Total | 320 | (100.0%) | 502 | (100.0%) |
| **Ethnicity** | | | | |
| Other | 80 | (25.2%) | 103 | (21.8%) |
| Makassarese \| iTaukei | 237 | (74.8%) | 369 | (78.2%) |
| Total | 317 | (100.0%) | 472 | (100.0%) |
| **Marital status** | | | | |
| Other | 44 | (13.9%) | 128 | (27.1%) |
| Married | 273 | (86.1%) | 344 | (72.9%) |
| Total | 317 | (100.0%) | 472 | (100.0%) |
| **Religion** | | | | |
| Other | 26 | (8.2%) | 74 | (15.7%) |
| Muslim \| Christian | 291 | (91.8%) | 396 | (84.3%) |
| Total | 317 | (100.0%) | 470 | (100.0%) |
| **Age** (years) | n | mean (SD) | n | mean (SD) |
| | 320 | 39.5 (12.2) | 502 | 42.2 (14.3) |

participation, with and without disabilities, and both genders) suggested that either their own low participation or that of others was because of a clash of RISE activities with other commitments, including looking after other people's children so they could participate instead and being occupied with studying or other activities. Another reason for low participation or engagement in Makassar was that the RISE activities were too long.

*There were some activities I wasn't involved in because I didn't have the time. However, my wife was involved and represented me. They also shared about the discussion with me.* (IDI, Man, Makassar).

Both Suva and Makassar residents said that some people did not participate because either they did not understand the purpose of RISE; or they were afraid their homes would be demolished by RISE; or they would be evicted. In Suva, some participants explained that some residents did not trust RISE because of a history of problematic development practices from other organizations:

*There are those who did not attend. They doubted RISE's project. This is because of the former work of other NGOs that came into this community; [it] was all about false promises.* (FGD, Man, Suva).

**Table 2. Parameter estimates from logistic regression models of socio-demographic predictors of participation in Makassar and Suva.**

| Variable | Odds Ratio (OR) | [95% CI] | p value |
|---|---|---|---|
| **Predictors of participation in Makassar participatory design workshops (N = 317)** | | | |
| Gender: (women vs. men) | 2.61 | 1.78,3.83 | <.001 |
| Marital status: (married vs. other status) | 2.14 | 1.34,3.42 | .002 |
| Ethnicity: (Makassarese vs. minority ethnic groups) | 1.85 | 1.00,3.43 | .052 |
| Age | 1.03 | 1.02,1.03 | <.001 |
| **Predictors of participation in Makassar household visits (N = 317)** | | | |
| Gender: (women vs. men) | 3.59 | 1.35,9.54 | .010 |
| Marital status: (married vs. other status) | 3.09 | 0.86,11.12 | .084 |
| Age | 1.05 | 1.00,1.09 | .035 |
| **Predictors of participation in Suva participatory design workshops (N = 499)** | | | |
| Disability: (disability vs. no disability) | 0.08 | 0.01,0.82 | .033 |
| **Predictors of participation in Suva household visits (N = 447)** | | | |
| Education: (secondary or above vs. primary or below) | 1.70 | 1.16,2.51 | .007 |
| Disability | 0.18 | 0.02,1.62 | .127 |

Makassar missing observations: 1 missing for participation in participatory design workshops, 2 missing for both ethnicity and marital status, 1 missing for participation in household visits. Suva missing observations: 3 missing from participation in participatory design workshops, 1 missing from disability, 3 missing from participation in household visits, 52 missing from education.

All models controlled for settlement-level clustering.

**Table 3. Barriers to participation in any RISE activities.**

| Reported barriers | Makassar | | Suva | |
|---|---|---|---|---|
| | n | (%) | n | (%) |
| Too busy with work, housework, or school | 29 | (80.6%) | 107 | (85.7%) |
| Did not want to participate | 2 | (5.6%) | 9 | (7.2%) |
| Not invited by RISE | 2 | (5.6%) | 6 | (4.8%) |
| Could not participate without assistance | 0 | (0.0%) | 1 | (0.8%) |
| Other | 3 | (8.3%) | 2 | (1.6%) |
| Total | 36 | (100.0%) | 125 | (100.0%) |

**4.3.b Barriers related to gender and social inclusion.** In addition to program-related barriers, participants also reported participation issues relating to their identities and social dynamics, including gender, disability, and poverty. Several participants noted that conflicting work responsibilities prevented men from participating in RISE group activities. Those participants reported that more men would participate if activities were held in the evenings or on weekends.

*I usually came to represent my family because my husband was working.* (IDI, Woman, Makassar)

In both Makassar and Fiji, it was also observed by the local program staff that more women than men participated in the larger scale formal activities, such as community workshops, and RISE was therefore thought of (especially in Makassar) as a 'women's program'. In both countries, it was thought that this perception was because RISE is a program about child health and water, which are usually more the domains of women than of men in Indonesia and Fiji. Additionally in Makassar, it was observed by some program staff and residents that this programmatic perception was also because many of the RISE staff who designed and led the activities were women. In Fiji, it was reported that the participatory design

workshops conflicted with men's relaxation time in evenings, which was in contrast to the expectations that men would participate more during evenings.

In both Makassar and Suva, some of the residents with mobility impairments said that their disability was why they had not participated in the community workshops to the extent they would have liked.

> *Actually, I want to go, ma'am but it is just my sore legs. So, we can't participate if we have problems.* (IDI, Woman, Makassar)

> *No, I did not attend because I am old. I only send my kids to attend this workshop. If I were physically fit I would really love to be part of the workshop.* (IDI, Woman, Suva)

In both countries, around half of the participants reported sending a family member to represent them if they were unable to attend (in Suva, RISE required that each household select one member to represent them at participatory design workshops due to COVID-19 restrictions); the majority of participants who said this had a disability. While most were satisfied with this arrangement, there was one Suva resident for whom this was not satisfactory because she lived alone (IDI, Woman, Suva).

One man living with a disability explained that a lot of RISE communication took place via cell phone, but those who could not afford one relied on the research team for information. A woman with a disability felt that those who were poor such as herself and those who lived near the edge of town (near the sea where the water table was too high for a septic tank) might miss out on the benefits of RISE:

> *Most people here work as fishermen. Sometimes some are lucky and sometimes some are not. We do not get the same income. Also, we live surrounded by water, so we can't dig [for a septic tank].* (IDI, Woman, Makassar)

### 4.4 What was the level of satisfaction among participants of primary RISE participatory design activities? (dose received)

In Makassar, when comparing preferred and experienced levels of influence among participants of at least one primary RISE participatory design activity, 161 (66%) of 245 were satisfied with their level of influence, 51 (21%) would have preferred more influence, and 33 (13%) would have preferred less influence (S4 Table). Among Suva participants of at least one primary RISE participatory design activity, 207 (70%) of 297 were satisfied with their level of influence, 27 (9%) would have preferred more influence, and 63 (21%) would have preferred less influence (S5 Table).

### 4.5 What were the socio-demographic predictors of satisfaction among participants of primary RISE participatory design activities? (dose received)

For Makassar participatory design participants, there were no significant predictors of influence discordance (either wanting more or less influence) over the primary participatory design process. For Suva participatory design participants, the odds of wanting *more* influence than what was experienced were 3.3 times higher for women compared to men (95% CI: 1.2, 8.8) (Table 4).

In Suva, the odds of wanting *less* influence than what was experienced were 2.6 times higher for religious minority respondents compared to Christian religious majority respondents (95% CI: 1.3, 5.1) and were 1.8 times higher for unmarried respondents (including single, divorced, or widowed) compared to married respondents (95% CI: 1.2, 2.9) (Table 4).

### 4.6 What were participants' opinions of RISE and engagement? (dose received)

There were varied opinions of the RISE program among the Makassar and Suva residents. In one Suva settlement, a resident explained that while she was happy with RISE, others there were experiencing participant fatigue with RISE: 'They

**Table 4. Parameter estimates from logistic regression models of socio-demographic predictors of influence discordance among Suva participatory design participants.**

| Variable | OR | [95 % CI] | | p value |
|---|---|---|---|---|
| **Predictor of wanting *more* influence (N = 235)** | | | | |
| Gender: women (vs. men) | 3.30 | 1.24, | 8.81 | .017 |
| **Predictors of wanting *less* influence (N = 260)** | | | | |
| Religion: religious minorities (vs. Christian majority) | 2.60 | 1.33, | 5.08 | .005 |
| Marital status: married (vs. other status) | 0.55 | 0.35, | 0.86 | .009 |

Models controlled for settlement-level clustering.

are tired of seeing [RISE] in their settlements' (IDI, Woman, Suva). However, almost all of the other Suva residents were positive about RISE:

*What RISE is currently doing, I am 100 percent thankful for.* (IDI, Woman, Suva)

Most of the participant groups cited the participatory and inclusive nature of the RISE program as being a mechanism that set it apart from other interventions that aim to improve life in informal settlements.
Most of the Makassar residents indicated that the RISE program was the first, in their experience, to directly include all residents in its activities:

*It is different because the RISE team involves the whole community.* (IDI, Woman, Makassar)

Residents in both countries pointed out that RISE was more inclusive than other programs they had experienced because there was no fee to participate whereas in other programs, in Makassar for example, the 'community had to pay if they wanted to attend' (IDI, Woman, Makassar). For the participants from at least one of the settlements in Suva, RISE was compared to another program that had requested each household invest money but never delivered the promised outputs:

*RISE brings in blessings from the outside but [the other program] takes away what we already have. This is the difference.* (FGD, Man, Suva)

There were mixed feelings about the level of engagement among the Suva participants. In one settlement, the men's FGD felt that consultation had been made with the settlement leaders, but not with the residents:

*They design it, they come, they present and they go, there has been no consultation. […] They did not take our opinion but I might be wrong, they might have taken it from the heads, from the committee members.* (FGD, Man, Suva)

In another Suva settlement, a young woman in a focus group explained that the youth there did not feel like their voices had been heard, including in the participatory design, but this was because 'the Chairman of the Committee thinks we are too young in age and experience to have our opinions heard.' (FGD, Woman, Suva)
However, in other Suva settlements, the residents felt like they had been sufficiently consulted.

*We all agreed about what they introduced during our community meeting. Their agenda was communicated and taken into our community meeting for people to agree to what was being introduced.* (FGD, Man, Suva)

Some of the Makassar, Australia, and USA-based staff explained that because the RISE program targeted households rather than individuals for participation in the activities, intra-household dynamics and power structures may have prevented individual household members from participating equitably in decision-making. For example, normally, decision-making power fell to whoever owned the land (which could be women or men), because RISE required land-owner approval to build infrastructure on household land. However, at other times, men or older residents had a greater influence over decision-making.

## 5 Discussion

In this mixed-methods process evaluation, we examined socio-demographic predictors of participation (reach) in community- and household-level participatory design activities within the context of the RISE intervention in Makassar, Indonesia and Suva, Fiji. We further assessed community members' level of satisfaction (dose received) with the participatory design activities and identified predictors of satisfaction. In Makassar and Suva, we observed that certain groups were disproportionately more likely to participate in both community- and household-level activities. In both countries, the majority of participants were satisfied with their level of influence in the RISE program, though some nuance around participant fatigue, distrust, and decision-making power was revealed in the qualitative data. In Makassar, satisfaction did not vary by socio-demographic characteristics. In Suva, we observed that women were more likely than men to want *more* influence; in contrast, religious minorities were more likely than the Christian religious majority to want *less* influence and unmarried people were more likely than married people to want *less* influence. This process evaluation highlights the successes and challenges of the intervention's participatory design phase and provides an example of how programs that aim for an inclusive participatory approach can evaluate their success in engaging diverse community members.

### 5.1 Intervention reach

Most groups of interest were reached through primary participatory activities, and household visits provided advantages to reach certain residents. In Makassar, women had high participation and were even more likely to report being reached through household visits than through community-level engagement. In Suva, household visits appeared to be more inclusive of people living with disabilities than community-level workshops, although this interpretation may be limited by sample size. These household-level findings build on a qualitative analysis of RISE's key mechanisms for achieving the gender and socially inclusive participatory approach for engaging diverse people, where household visits were reported to engage those who could not participate in the larger community venues due to socio-demographic characteristics, marginalization, or time conflicts [27]. In other studies, household visits have been observed to be an important activity to reach caregivers and women participants [6,33].

The findings indicate that household visits had both benefits and consequences for inclusion. While both participants and RISE staff reported that household visits were an important activity to reach groups of interest, RISE staff noted that hierarchical intra-household dynamics, along with project requirements for landowner approval of final decisions related to infrastructure, may have limited who was involved in decision-making. Additionally, Makassar RISE staff said that men had *higher* participation in household visits than what was observed in the quantitative results, because even if men were not present for participatory design activities, they still typically made the final design decisions for their households (although this varied by who the landowner was). This aligns with a systematic review of women's empowerment and water and sanitation, which identified that women often have been excluded from household-level decision-making for water and sanitation infrastructure [11]. Strategies to mitigate inequitable intra-household dynamics that limit decision-making power can include collecting participant-level data to inform infrastructure design, or engaging with informal or established groups, like women's and savings groups or organizations of persons with disabilities, to better include participants [27,35]. Additionally, given that landownership had a large influence on who made final RISE household infrastructure

decisions, interventions operating on private household land need to acknowledge this power dynamic and facilitate shared decision-making within households.

## 5.2 Barriers to participation

For residents who did not report participating in any RISE design activity, timing conflicts were the most common barrier to participation. Other assessments of WASH program delivery found similar challenges with scheduling community-level events [6]. Adapting to when diverse groups of people are available, not just when the most people can attend, is an approach to include a broader array of perspectives in participatory design [42]. Residents also discussed barriers to participation relating to social exclusion, like mobility disability. Assessing workshop facilities for accessibility and designing appropriate communication methods are steps that programs can use to accommodate participants living with disabilities. In addition, a review of disability measures recommended using multiple, valid disability self-assessments (like the Washington Group measure and the Katz's Activities of Daily Living Index) to measure people's functionality [43]. These results further highlight the value of assessing social and structural barriers before and during implementation of participatory design processes.

## 5.3 Participant satisfaction and influence (dose received)

The qualitative dose received findings suggest that participants were generally satisfied with RISE's participatory approach, including deliberately involving all residents and fee-free participation. Residents also distinguished RISE from earlier external projects that did not deliver expected outcomes, which echoes a structural distrust with development work [34,44]. Indeed, some residents reported not trusting RISE because they did not understand the project's purpose. A similar feeling of distrust was found by a study on the effect of RISE participatory design on social capital, which identified that over half of Suva intervention communities experienced conflict with RISE over COVID-19-related food distributions and the representativeness of RISE liaison committees [35]. These findings highlight the critical importance of trust-building in community development projects, including discussing community priorities and expectations. Additional approaches to build trust with communities include adapting to community needs and establishing regular contact with participants, both of which require programs to allocate sufficient time and resources [27].

While most participants in both countries were satisfied with their level of influence over decision-making, the quantitative results also suggest that social dynamics around gender, religion, and marital status affected participant satisfaction. Women in Suva were more likely to report wanting *more* influence over RISE compared to men, despite having comparable participation. Patriarchal gender norms may also be reflected in this discrepancy, as RISE staff observed that most women tended to follow the decisions of husbands or male heads of households across Suva communities [45,46]. Informal engagement channels and gender norms reinforced by RISE staff also may have had a role with this satisfaction finding: after the formal participatory design workshops ended, RISE staff socialized with primarily men during *kava* drinking sessions, an important tradition in Fiji, that was a key time for further discussion of RISE. Women who attended the participatory design workshops, but did not attend the *kava* sessions because of other evening responsibilities, may have felt less influential as a result. Inequitable decision-making also has been reported in water and sanitation design and planning, where women and other socially excluded groups felt like their voices were not heard during intervention meetings, despite being considered key informants by practitioners [4].

In contrast to those who wanted more influence, religious minorities in Suva were more likely to report wanting *less* influence over RISE compared to the Christian religious majority. Hindu participants, in particular, may have wanted to avoid conflict during RISE decision-making because of inter-ethnic tension between Indo-Fijians (majority Hindu) and iTaukei indigenous Fijians (majority Christian) that has evolved since the British colonization of Fiji [47,48]. Salinger et al.'s study on RISE participatory design and social capital found similar tensions in ethnically heterogeneous Suva settlements, where some iTaukei residents were critical of a perceived lack of Indo-Fijian participation [35]. Importantly, RISE staff

noted that Indo-Fijian engagement increased with additional, intentional outreach. This finding further supports Francis et al.'s recommendation of having a diverse team to engage diverse participants during participatory design [27]. Given how social dynamics can affect participants' decision-making, assessing participant influence can be particularly relevant for participatory design processes to see if approaches are promoting inclusive community engagement and how such processes may be improved.

Participant-reported influence is an important addition to Saunders et al.'s framework definition of dose received, which is the extent of community satisfaction and the extent to which participants engage with intervention components [30]. In practice, WASH process evaluations have also conceptualized dose received as participants' acceptability and awareness of intervention activities [32–34]. Further, a systematic review of health promotion studies found that dose received has been used to assess attendance, completion of activities, and use of materials among other applications [49]. Our process evaluation expanded Saunders et al.'s definition of dose received beyond attendance or awareness of an intervention to measure participants' self-reported influence over decision-making. Doing so helped capture community members' engagement with RISE participatory design activities beyond broad satisfaction or intervention receipt. The integration of qualitative approaches helped us to further understand participants' opinions of RISE from their own perspective.

## 5.4 Implications for programs and policy

Our process evaluation findings have implications for future programming with participatory design, including the upcoming tranche of RISE control group settlements. First, our results further support the importance of understanding existing community and intra-household dynamics in order to support inclusion and equitable decision-making [35]. Approaches include fostering partnerships with existing community groups (including women's groups, savings groups, and organizations of persons with disabilities), collecting individual-level data for design options, and fostering shared household decision-making [27]. While final decisions may still rest with a landowner or head of household, these recommended approaches may be valuable for bringing additional voices and perspectives into decision-making processes. Similarly, it is recommended that programs spend time and resources before starting participatory design processes to understand whether and how diverse community members would like to participate, as people will have varying degrees of interest in engagement [42]. During the design process, it is advised that implementers be cognizant of informal engagement methods and who may be included and excluded from those spaces. Lastly, mixed-methods, framework-guided process evaluations are valuable to assess how participatory design is implemented, with both quantitative and qualitative data assessing reach and satisfaction.

## 6 Strengths & limitations

This process evaluation has a number of strengths. The research was nested within the larger RISE randomized control trial in intervention settlements. Given this, survey enumerators had prior experience with RISE data collection methods and with community members, and household contact information was available in Makassar from earlier phone surveys. In addition to data collector experience, many community members also had familiarity with RISE enumerators prior to data collection. Qualitative data collection sought the emic perspective from participants with varied socio-demographic characteristics across study locations and intentionally included people who were marginalized [27]. Finally, the research was able to build on rich findings from additional studies about RISE's participatory design phase.

Several limitations also exist for this process evaluation. This study assessed the formal participatory design phase, but the participatory design process has since continued throughout infrastructure construction. As the process evaluation was a secondary analysis of the quantitative and qualitative data, analysis was limited to two process indicators. During quantitative data collection, sampling bias may have occurred, as those who participated in participatory design activities may have been more likely to respond to this study's survey. Recall bias also may have occurred, as the Makassar survey was administered 11–13 months after the end of RISE's participatory design phase. Data collectors for the quantitative

and qualitative data may have introduced reporting bias; while they were not directly involved in RISE implementation, they were affiliated with the larger RISE trial. While gender inclusion was a key research interest, the qualitative data collection did not specifically recruit those who were non-binary or transgender [27]. Data collection overall was only conducted with adults, which is a limitation given that qualitative data from Suva indicated that some youth felt excluded from the participatory process.

## 7 Conclusion

This mixed-methods process evaluation found that RISE participatory design workshops and household visits achieved good reach and dose received. The study also identified gender and social inequities in design participation and influence over decision-making. We encourage future WASH participatory design processes to develop meaningful engagement with communities and local organizations to better involve people of interest. Process evaluations are recommended to further understand and promote community engagement during and following participatory design.

## Supporting information

**S1 Table. Survey questions and response options for RISE participation, barriers to participation, experienced influence, and preferred influence.**
(DOCX)

**S2 Table. Qualitative in-depth interview and focus group discussion participant demographics.**
(DOCX)

**S3 Table. Suva and Makassar participation in participatory design workshops and household visits by socio-demographic characteristics.**
(DOCX)

**S4 Table. Experienced and preferred influence among primary participatory design participants in Makassar.**
(DOCX)

**S5 Table. Experienced and preferred influence among primary participatory design participants in Suva.**
(DOCX)

## Acknowledgments

We are grateful to all RISE project participants for sharing their time and participation in the research. We would like to thank Fiona Barker, Data Manager and Chief Investigator on the RISE trial, for the management of all quantitative data used in our study; Karin Leder, Director of Research on the RISE trial; Soropepeli Ramacake of the University of the South Pacific for assisting with data collection coordination in Fiji; Isoa Vakarewa for his project administration in Fiji; Ruzka Taruc for her project administration in Indonesia; and Arantxa Bonifaz Rosas of Emory University for her review of transcripts. We are thankful to our research partners at the University of the South Pacific including Litea Meo-Sewabu for research coordination and supervision; Camari Koto, Sunia Baikeirewa, and Malakai Waqa for qualitative data collection in Suva; and Natasha Khan for translations. We thank Inoke Droya, Jese Cabenalevu, Misila Nasolo, Saliman Bibi, Maraia Luveniyali, and Evelyn Chand, the volunteers who shared their time as data collectors and transcribers in Suva. We thank Sudirman Nasir, our research partner at Universitas Hasanuddin, and Rafika Ramli, Amanda Pricella Putri, Betrin Natasya, and Rachma Rahim, who contributed to the study as transcribers and translators in Makassar. We would also like to thank the following contributors for their assistance with interpretation of study results: Fitriyanty Awaluddin, Adrianto Hidayat, Raniyah Muhammed, Savu Nofoimuli, Bulou Ratulevu, Autiko Tela, Josaia Thaggard, and Iliesa Wise. This study

was completed as part of the Revitalising Informal Settlements and their Environments (RISE) program (https://www.rise-program.org/) on behalf of the RISE Consortium (doi.org/10.26180/ctjf-vf69).

## Author contributions

**Conceptualization:** Thea L. Mink, Allison P. Salinger, Sheela S. Sinharoy.

**Data curation:** Allison P. Salinger, Naomi Francis.

**Formal analysis:** Thea L. Mink, Naomi Francis.

**Funding acquisition:** Becky Batagol, Michaela F. Prescott, Sheela S. Sinharoy.

**Investigation:** Naomi Francis, Becky Batagol.

**Methodology:** Allison P. Salinger, Naomi Francis, Sheela S. Sinharoy.

**Project administration:** Becky Batagol, Sheela S. Sinharoy.

**Supervision:** Allison P. Salinger, Becky Batagol, Sheela S. Sinharoy.

**Visualization:** Thea L. Mink, Naomi Francis.

**Writing – original draft:** Thea L. Mink, Naomi Francis.

**Writing – review & editing:** Allison P. Salinger, Naomi Francis, Becky Batagol, Kerrie Burge, Noor Ilhamsyah, Losalini Malumu, Liza Marzaman, Michaela F. Prescott, Nur Intan Putri, Sheela S. Sinharoy.

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
