## [Decision Letter · Decision Letter 0]

PGPH-D-24-01162

Who participates in ‘participatory design’ of WASH infrastructure: a mixed-methods process evaluation

Dear Dr. Sinharoy,

Thank you for submitting your manuscript to PLOS Global Public Health. After careful consideration, we feel that it has merit but does not fully meet PLOS Global Public Health’s publication criteria as it currently stands. Therefore, we invite you to submit a revised version of the manuscript that addresses the points raised during the review process.

We look forward to receiving your revised manuscript.

Kind regards,

Rajiv Sarkar

Academic Editor

Additional Editor Comments (if provided):

1. Please provide a summary (with timelines, if possible) of the different Community engagement activities conducted as part of the  RISE trial. This can be included as a figure & will help contextualize the findings presented in this manuscript.

2. The findings of the multivariable logistic regression analysis (Tables 2, 4 & 5) can be combined into a single table. If the analysis was conducted separately for Makassar and Suva, please present the results separately, either within the main text or as supplementary material.

Reviewers' comments:

Reviewer's Responses to Questions

**Comments to the Author**

1. Does this manuscript meet PLOS Global Public Health’s publication criteria ? Is the manuscript technically sound, and do the data support the conclusions? The manuscript must describe methodologically and ethically rigorous research with conclusions that are appropriately drawn based on the data presented.

Reviewer #1: Yes

Reviewer #2: Yes

2. Has the statistical analysis been performed appropriately and rigorously?

Reviewer #1: Yes

Reviewer #2: Yes

3. Have the authors made all data underlying the findings in their manuscript fully available (please refer to the Data Availability Statement at the start of the manuscript PDF file)?

Reviewer #1: Yes

Reviewer #2: Yes

4. Is the manuscript presented in an intelligible fashion and written in standard English?

Reviewer #1: Yes

Reviewer #2: Yes

5. Review Comments to the Author

Reviewer #1: Reviewer Comments

Title: Who participates in ‘participatory design’ of WASH infrastructure: a mixed-methods process evaluation

Summary

The authors aimed to assess reach and satisfaction of WASH interventions in selected communities. The article is well written and spells out clearly the research questions and research process. The methodology is rigorous and justifiable. The paper also makes informed recommendations for future programming.

Major Issues

No major issues of concern were identified.

Minor Issues

The authors, in the bid to assess dose received asked all study participants “how much influence respondents and their households felt 142 they had over RISE decision-making, along with how much influence respondents and their 143 households would have preferred”. These questions were asked regardless of whether subjects participated in the RISE intervention or not. This begs the question why study participants who did not contribute to the intervention were still included. Was this to find out their preferred influence had they participated. If this is so, then must be clarified. Furthermore, how will you be able to classify this group then as matched or discordant, since the ‘experienced’ component is unavailable to do the comparison the authors describe.

The authors could also clarify why the p-value significant level was unconventionally set at 0.25.

Reviewer #2: The manuscript described a mixed-method approach to evaluate the participation by community members of RISE programs in two countries, with the intention to identify factors and barriers affecting participation. The report on this process evaluation is interesting and of importance, but several clarifications could be made to facilitate reader’s understanding of the context.

1. To provide a background and context to the study, please give a description of the following concerns so that readers may get a general idea about the motivation to participate.

i. What is the relative importance of this project among similar development programs in recent years?

ii. What is the relationship between the agents of implementation and agents conducting the participatory processes?

iii. What is the perception of local residents regarding the two questions above? What would motivate people to participate?

iv. Please report on local residents’ interest in the RISE program and perceived need for a water-sanitary service?

2. Related to the previous comment, the authors did not explain how the RISE program is going to have impact on the use of land, and how this was communicated with the residents before?

3. In assessing reach (p. 6): “We operationalized reach as the proportion of quantitative survey respondents reporting participation in two primary RISE participatory design activities.” This would not be satisfactory if the survey procedure shares the same limitations in reaching the population (such as those who have to work, or those with different language preferences). It will be helful to report the strategies of recruitment for the survey, how it is different from invitation to participate, and the participation rate of the survey (as a proportion of the total number of intended/targeted population).

4. The authors reported (p. 15) that “For Suva participatory design workshops, respondents living with disabilities were 92% less likely to report participating compared to respondents without disabilities.” It would mean that there is a super high rate of disability in Suva; please check for accuracy.

5. In assessing the community member’s level of satisfaction (dose), is it possible to separate into steps, such as information received before the meeting, clarity about the agenda/purpose of the meeting, clarity of information during the meeting, and the arrangement of participation (the rules of the proceeding was clear and encouraging participation), etc.?

6. Please give examples of how the participants may perceive that they have had “influenced on decision-making” (i.e., which decisions, who made the decision, how was the decision made [e.g., was there voting], was the influence observable during the consultation). It is especially difficult to understand those who “would have preferred less influence”, without given the context. For example, they may have chosen not to attend, or not to speak during the meeting.

6. PLOS authors have the option to publish the peer review history of their article (what does this mean? ). If published, this will include your full peer review and any attached files.

**Do you want your identity to be public for this peer review?** For information about this choice, including consent withdrawal, please see our Privacy Policy .

Reviewer #1: No

Reviewer #2: No

---

## [Decision Letter · Decision Letter 1]

PGPH-D-24-01162R1

Who participates in ‘participatory design’ of WASH infrastructure: a mixed-methods process evaluation

Dear Dr. Sinharoy,

Thank you for submitting your manuscript to PLOS Global Public Health. After careful consideration, we feel that it has merit but does not fully meet PLOS Global Public Health’s publication criteria as it currently stands. Therefore, we invite you to submit a revised version of the manuscript that addresses the points raised during the review process.

We look forward to receiving your revised manuscript.

Kind regards,

Rajiv Sarkar

Academic Editor

Journal Requirements:

Additional Editor Comments (if provided):

In addition to the minor comments from Reviewer 1, please also address the following comments:

Were the data collectors for the quantitative surveys and the qualitative interviews involved with trial implementation? If yes, could their involvement have introduced reporting bias?Tables 1 and 3: Please present the percentages (%) in parenthesis. Also, for Table 1, kindly highlight that the age was in completed years.Tables 2 and 4: For ease of reading, please separate the lower & upper bound confidence intervals with a coma (,).Page 16: “For every one year increase in age in years, the adjusted odds of participation increased by 3% (95% CI: 1.0, 1.0).” The upper-bound confidence interval, as presented gives a false impression that it is same as the lower-bound confidence interval (possibly due to rounding) – kindly review & update, as needed.

Reviewers' comments:

Reviewer's Responses to Questions

**Comments to the Author**

1. If the authors have adequately addressed your comments raised in a previous round of review and you feel that this manuscript is now acceptable for publication, you may indicate that here to bypass the “Comments to the Author” section, enter your conflict of interest statement in the “Confidential to Editor” section, and submit your "Accept" recommendation.

Reviewer #1: All comments have been addressed

Reviewer #2: All comments have been addressed

2. Does this manuscript meet PLOS Global Public Health’s publication criteria ? Is the manuscript technically sound, and do the data support the conclusions? The manuscript must describe methodologically and ethically rigorous research with conclusions that are appropriately drawn based on the data presented.

Reviewer #1: Yes

Reviewer #2: Yes

3. Has the statistical analysis been performed appropriately and rigorously?

Reviewer #1: Yes

Reviewer #2: Yes

4. Have the authors made all data underlying the findings in their manuscript fully available (please refer to the Data Availability Statement at the start of the manuscript PDF file)?

Reviewer #1: Yes

Reviewer #2: Yes

5. Is the manuscript presented in an intelligible fashion and written in standard English?

Reviewer #1: Yes

Reviewer #2: Yes

6. Review Comments to the Author

Reviewer #1: Paper Review

Who participates in ‘participatory design’ of WASH infrastructure: a mixed-methods process evaluation

31 Is the one billion people exact? You may say approximately; also as you are referring to people use ‘who’ instead of ‘which’. This is the opening sentence of your work so should not attract any ambiguity.

83 Consider ‘few’ in place of ‘handful’

Clear methodology and good analyses from results

Overall, an impressive addition to knowledge

Reviewer #2: The authors have responded to my comments, some of them by referring to a previous manuscript published by the research team.

7. PLOS authors have the option to publish the peer review history of their article (what does this mean? ). If published, this will include your full peer review and any attached files.

**Do you want your identity to be public for this peer review?** For information about this choice, including consent withdrawal, please see our Privacy Policy .

Reviewer #1: **Yes: ** Dr Dr Nana Mireku-Gyimah

Reviewer #2: No

---

## [Editor Report · Decision Letter 2]

Who participates in ‘participatory design’ of WASH infrastructure: a mixed-methods process evaluation

PGPH-D-24-01162R2

Dear Dr. Sinharoy,

We are pleased to inform you that your manuscript 'Who participates in ‘participatory design’ of WASH infrastructure: a mixed-methods process evaluation' has been provisionally accepted for publication in PLOS Global Public Health.

Best regards,

Rajiv Sarkar

Academic Editor